Genome-wide identification and functional characterization of CDPK gene family reveal their involvement in response to drought stress in Gossypium barbadense

Shi Guangzhen
Zhu Xinxia zhuxxshz@126.com 302641316@qq.com
Key Laboratory of Oasis Town and Mountain-basin System Ecology of Xinjiang Production and Construction Corps, College of Life Sciences, Shihezi University , Shihezi , Xinjiang , China
Atif Rana Muhammad
Electronic publication date: 2022 Feb 8
Publication date: 2022
Volume: 10
Electronic Location ID: e12883
Received 2021 Nov 4; Accepted 2022 Jan 13
Copyright: ©2022 Shi and Zhu
Copyright year: 2022
Copyright holder: Shi and Zhu
License: This is an open access article distributed under the terms of the Creative Commons Attribution License, which permits unrestricted use, distribution, reproduction and adaptation in any medium and for any purpose provided that it is properly attributed. For attribution, the original author(s), title, publication source (PeerJ) and either DOI or URL of the article must be cited.
License URL: https://creativecommons.org/licenses/by/4.0/

Keywords: Identification, CDPKs, Drought, Stress, Gossypium barbadense

Funding: National Natural Science Foundation of China 31760066 This work was supported by the National Natural Science Foundation of China (No. 31760066). The funders had no role in study design, data collection and analysis, decision to publish, or preparation of the manuscript.

==============================
Background

Calcium dependent protein kinases (CDPKs) are a class of important calcium signal sensing response proteins, which play an important regulatory role in response to abiotic stress. However, researchers have not been excavated CDPKs’ role in drought in sea-island cotton(Gossypium barbadense L. ‘H7124’).

Results

Eighty-four CDPK genes have been identified in G. barbadense. These GbCDPK genes are unevenly distributed on 26 chromosomes, and segmental duplication is the significant way for the extension of CDPK family. Also, members within the same subfamily share a similar gene structure and motif composition. There are a large number of cis-elements involved in plant growth and response to stresses in the promoter regions of GbCDPKs. Additionally, these GbCDPKs show differential expression patterns in cotton tissues. The transcription levels of most genes were markedly altered in cotton under heat, cold, salt and PEG treatments, while the expressions of some GbCDPKs were induced in cotton under drought stress. Among these drought-induced genes, we selected GbCDPK32, GbCDPK68, GbCDPK74, GbCDPK80 and GbCDPK83 for further functional characterization by virus-induced gene silencing (VIGS) method.

Conclusions

In conclusion, the principal findings of this prospective study are that CDPKs were associated with drought. These findings provide a solid foundation for the development of future molecular mechanism in sea-island cotton.

Introduction

Cotton is a significant financial and oil crop in the world, which provides the most commonly natural fiber materials for cotton textile industry. Hence, cotton has been developed for a huge scope around the world. Xinjiang is the largest cotton-growing region in China and even in the world. Sea-island cotton is the second-largest cotton planted in the world, which has shown in comparable superiority for its highly desirable fiber properties (Yu et al., 2021), which is grown in large areas in southern Xinjiang, where it is arid, with long sunlight and little rainfall. The growing area of sea-island cotton is limited due to the relatively low lint yield and narrow adaptation. Drought is one of the most serious climatic disasters affecting cotton quality and production (Ault, 2020). This is a problem that needs to be solved urgently.

Calcium dependent protein kinases are a class of Ser/Thr type protein kinases, which are usually composed of four core domains: N-terminal variable domain, protein kinase domain, autoinhibitory domain and C-terminal calmodulin like domain (Cheng et al., 2002). Plants sense environmental stimuli and transmit extracellular signals into intracellular through typical mechanisms and cause a series of responses. The C-terminal calmodulin like domain is composed of a pair of helix loop EF-hands structure. The combination of Ca2+ and EF-hands structure can cause the change of the spatial structure of CDPKs, which leads to the release of the inhibition of the self inhibitory domain, and promotes the recovery of the protein kinase activity of CDPKs, phosphorylates the downstream regulatory factors, and transmits the signal to the downstream regulatory network (Shi et al., 2018).

When plants are faced with abiotic stresses, CDPKs can respond to abiotic stresses and improve plant resistance to abiotic stresses by regulating the specific expression of genes. Overexpression of AtCPK4, AtCPK8, AtCPK10 and AtCPK11 can significantly improve the drought resistance of transgenic plants (Zou et al., 2010; Zhu et al., 2007; Zou et al., 2015). It is reported that OsCDPK4 plays an important role in salt tolerance and drought stress of rice (Campo et al., 2014). Overexpression of OsCDPK9 play an active role in drought stress tolerance (Wei et al., 2014). Overexpression of tobacco ZoCDPK1 gene can affect salt and drought stress (Vivek et al., 2017). Overexpression of ZmCPK4 in the transgenic Arabidopsis enhanced drought stress tolerance (Jiang et al., 2013).

Although CDPKs genome-wide identification and functional studies have been carried out in Arabidopsis (34) (Hrabak et al., 2003), rice (31) (Ray et al., 2007) and maize (31) (Kong et al., 2013), there is no report on CDPK gene family in G. barbadense. Based on G. barbadense genomes, we used bioinformatics to identify the members of CDPK gene family and predict the molecular structure, physiological characteristics of the protein. Analyzing the chromosome location, evolution and classification, the expression profiles of the CDPK gene. Moreover, we further investigated the function of the selected CDPKs in G. barbadense defense against drought stress by using virus-induced gene silencing (VIGS) technology. The results of this experiment provide an important theoretical basis for further understanding of the evolution and function of the CDPK and the cultivation of drought-tolerant cotton varieties in G. barbadense.

Materials & Methods

Identification of the CDPK family genes in Gossypium barbadense

The G. barbadense genome data comes from https://cottonfgd.org/ (Zhu et al., 2017), The Arabidopsis CDPK protein sequences of family members were downloaded from the Arabidopsis genome database TAIR website (https://www.arabidopsis.org/). Using Pfam online database (http://pfam.xfam.org/) to download the CDPK gene seed file EF-hand (Pfam ID: PF00036) and serine/threonine protein kinase (Pfam ID: PF07714). The hmmsearch program of HMMER3.0 was used to identify protein sequences containing CDPKs conserved domains. All CDPK protein sequences were extracted NCBI CDD and Pfam online database were used to verify the conservative domain. Finally, all members of CDPK gene family were obtained and named according to their different positions on chromosomes in G. barbadense.

Molecular structure and physicochemical properties

The Protparam tool (https://web.expasy.org/protparam/) was used to predict the molecular weight, isoelectric point and other relevant information of the CDPK protein. Using the online tool myristoylation (https://web.expasy.org/myristoylator/) predict N-myristoylation, and palmitoylation prediction tools is CSS-plam. Subcellular localization of the CDPK proteins were predicted using an online tool wolf POSRT (https://www.genscript.com/psort.html).

Phylogenetic tree, gene structure, conserved motif, and promoter region analyses

The ClustalX software was used to perform multiple sequence alignments on the CDPK proteins of G. barbadense and Arabidopsis, and the phylogenetic tree was constructed by MEGAX software: the Neighbor-Joining algorithm was used, and the Bootstrap repeat value was 1,000 times (Kumar et al., 2018). The online GSDS tool (http://gsds.cbi.pku.edu.cn) (Hu et al., 2015) was used to visualize the exon-intron structures of the CDPK genes. Using MEME (http://meme-suite.org/tools/meme) (Bailey et al., 2009) online software to analyze the conserved motifs of G. barbadense CDPK, the parameters were set as the maximum number of motifs found was 10, and the longest motif length was 100 nt (Nucleotide). The 1,500 bp DNA sequence upstream was extracted, and the PlantCARE (Lescot et al., 2002) database (http://bioinformatics.psb.ugent.be/webtools/plantcare/html/) was used to predict the possible cis-acting elements.

Chromosome location and gene replication

The information of CDPK gene family chromosome physical location was extracted from G. barbadense genome annotation file, and the chromosome physical distribution map of the genes was drawn with MapInspect. The definition of gene duplication is as follows: (1) the sequence length is greater than 80% of the gene; (2) the identity of alignment region is greater than 80%; (3) only one repeat event is calculated for closely linked genes. According to the chromosomal location of CDPK gene, tandem duplication and segment duplication can be identified. The KaKs_Calculator2.0 of Tbtools (Chen et al., 2020) was used to calculate the non-synonymous mutation rate (Ka) and the synonymous mutation rate (Ks) of the replication gene pair in G. barbadense.

Expression profile analysis, RNA extraction and real-time PCR

The transcriptome data of GbCDPK genes in the tissues and the abiotic stresses were from Sequence read archive, SRA, and the accession number was PRJNA274882. The RPKM value (the reads per kilobase of transcript per million mapped reads) represents the abundance of gene expression at the transcription level. Using Tbtools (Chen et al., 2020) map the gene expression profiles of different tissues and different abiotic stresses. Total RNA is extracted with the plant total RNA extraction kit (TIANGEN, Beijing, China). RNA reverse transcription adopts HiScript II QRT SuperMix for qPCR (R222-01), a product of Nanjing Novazin. The cDNA was used as a template to perform RT-PCR and qPCR experiments. The internal reference gene is the ubiquitinated protein Ubiquitin7 (UBQ7), and the RT-PCR reaction uses 1.0% agarose gel to detect the amplification results. The quantitative instrument uses Roche products, and the calculated Ct value is used to calculate the relative expression of the gene. The calculation method is 2−ΔΔCt (Pfaffl, 2001). Each gene amplification result contains at least 3 technical replicates and 3 biological replicates. The amplification system and procedure are shown in Appendix A and Appendix B.

Experiment treatment

The G. barbadense was grown for 4 weeks in a high humidity and high nutrient environment (nutritional soil: vermiculite: perlite = 3:1:1) at 28 °C, and then treated with abiotic stress to well growing cotton seedlings. Cold stress and high temperature stress are moved plants to incubators at 4 °C and 37 °C, respectively. For salt stress, seedlings are irrigated with 200 mM NaCl. Seedlings are irrigated with 20% PEG6000 for drought stress. One hour after the start of the treatment, the second true leaf of the treated cotton seedling was harvested and stored in a refrigerator at −80 °C. Each treatment included 3 seedlings, and each treatment had three experimental replicates.

VIGS experiment in G. barbadense

The silenced fragments of each CDPK gene were amplified from cotton and inserted into VIGS vector (TRV-00) to generate CDPK silent constructs. TRV-00 without the silent segment was used as control. Consequently, the vector was transformed into the GV3101. GV3101 contained the same amount of TRV-00 and TRV-CDPK, and infiltrated into the cotyledons of 14-days-old cotton seedlings by syringe infiltration method. Two weeks after inoculation, the silenced cotton seedlings were used for gene expression analysis and identification PEG treatment.

Methods of physiological index

RWC was measured by using the formula: RWC%=FW−DWTW−DW×100. Where FW, stands for fresh weight TW for turgid weight and DW for dry weight (Loutfy et al., 2012). G. barbadense leaves were soaked in 30 mL deionized water at 25 °C and then shaken for 30 min. Heated the solution to 100 °C for 30 min and then at room temperature. Measure REL with conductivity meter (Yang et al., 2012). The content of MDA, Pro, soluble sugar measured using the fresh sample. The amount of MDA was measured by the thiobarbituric acid method (Heath & Packer, 1968). The content of Pro was determined by the Sulfosalicylic acid indandione method (Bates, Waldren & Teare, 1973). Soluble sugar content was measured as described by the anthrone method and determined at 620 nm by a spectrophotometer (Fales, 1951). Water loss rate (%) = W1−W2W1×100. W1 stands for weight of detached blade before water loss. W2 weight of detached blade after 1, 2, 3, 4 and 5 h of water loss (Czyczyło-Mysza et al., 2018). All physiological results were repeated three times and a one-way analysis of variance (ANOVA) was performed with Prism to detect significant differences between both treatments (P < 0.05, P < 0.01).

Results

Identification and proteins characteristic of CDPKs in Gossypium barbadense

CDPK protein sequences of Arabidopsis were provided as direct queries against the G. barbadense genome to perform BLASTP and hidden Markov Model (HMM) to identify the GbCDPK gene family members. After NCBI-CDD and Pfam software database verification, 84 CDPK genes were identified in G. barbadense genome (Table S1).

The biochemical characteristics of the CDPK gene, such as amino acid sequence, molecular weight and isoelectric point were analyzed by the ProtParamtool online software. The results show that the CDPK genes are conservative. The longest CDPK protein encodes 648 amino acids, and the shortest only encodes 155 amino acids. The molecular weight of GbCDPKs varies 17.99 KDa to 71.854 KDa, and the theoretical isoelectric point ranges from 4.313 to 9.481. Multiple sequence alignment analysis indicated that all of GbCDPKs have four core domains. In addition, all the GbCDPKs have EF-hands motifs, which allows binding of Ca2+. It reported that 71 GbCDPK proteins contain four EF-hands, nine proteins (GbCDPK25, GbCDPK26, GbCDPK39, GbCDPK49, GbCDPK53, GbCDPK67, GbCDPK68, GbCDPK77, GbCDPK81) have three EF-hands, and remaining four proteins (GbCDPK22, GbCDPK38, GbCDPK61, GbCDPK64) only contains 2 EF-hands. Our bioinformatic analysis revealed that 40 members harbored potential N-myristoylation sites, and 74 GbCDPK proteins contained S-palmitoylation sites (Table S1). Using TMHMMServerv2.0 prediction, it was found that none of the 84 CDPK proteins had a transmembrane structure. Using Wolf POSRT tool to predict subcellular localization, it was found that 22 CDPK proteins are in the chloroplast, 33 CDPK proteins are in the cytoplasm, eight are in the nucleus, and the others are in the mitochondria, extracellular, vacuole, endoplasmic reticulum, and peroxisome.

Phylogenetic analysis of GbCDPKs

To assess the phylogenetic relationship of CDPK protein in G. barbadense. A total of 118 CDPK proteins from G. barbadense and Arabidopsis were used to construct the phylogenetic tree by adjacency method (Fig. 1). According to the results of phylogenetic tree clustering, 84 CDPK genes can be divided into four subfamilies. Referring to the research results of Arabidopsis, four subfamilies were named as class I, class II, class III, and class IV. Further analysis of the clustering results of CDPK members revealed that the gene structures in the homologous gene pair were similar. The number of subgroup II proteins was the largest, including 30 G. barbadense and 10 Arabidopsis CDPK proteins, accounting for 35.71% and 29.41% of the total, respectively. The first subgroup includes 27 CDPK proteins from G. barbadense and 10 CDPK proteins from Arabidopsis. The third subgroup includes 19 CDPK proteins from G. barbadense and eight CDPK proteins from Arabidopsis. The fourth subgroup has the least number, including only eight CDPK proteins from G. barbadense and 3 CDPK proteins from Arabidopsis, indicating that it’s far from each other.

Figure 1 Evolutionary relationship of CDPK Family in G. barbadense.

Chromosomal distribution and gene duplication

To investigate the chromosomal locations, chromosomal maps were drawn for the 84 CDPK genes from G. barbadense by MapInspect software. All 84 GbCDPK genes are mapped on 13 pairs of chromosomes (Fig. 2). Furthermore, cotton genome includes A-subgroup and D-subgroup that all contain 42 GbCDPKs, respectively, and the duplication events may illuminate the mechanism about the expansion of GbCDPK gene family. There are six genes on chromosome 4 and 13 of A-subgroup, five genes on chromosome 2 of A-subgroup and 5 and 13 of D-subgroup, respectively. In contrast, there is only one gene on chromosome 3 and 8 of A-subgroup and chromosome 8 of D-subgroup, and there are 2–4 genes on other chromosomes. These results indicated that CDPK genes were widely distributed in G. barbadense. In order to speculate the possible relationship between GbCDPK genes and potential gene replication in the G. barbadense genomes, we analyzed the occurrence of tandem duplication and large-scale segmental replication during the evolution of the gene family (Table S2). Among 84 genes, 41 pairs of homologous genes were found in phylogenetic tree (Fig. 1). There were 40 pairs of segmental replications and one pair of tandem replications (GbCDPK10/GbCDPK11).

Figure 2 Distribution of CDPK gene family on chromosomes.

To study the evolutionary replication relationship of genes in G. barbadense on chromosome segments, MCscanx software was used to detect the replication genes. 12, 16 and 60 pairs of whole genome replicates were detected among A, D and AD subgroups in G. barbadense, respectively (Fig. 2). KaKs_Calculator was used to perform Ka/Ks analysis replication gene pairs. It was found that the Ka/Ks of all gene pairs were less than 1, indicating that the CDPK gene of G. barbadense might have undergone strict purification and selection during the evolution process, implying that the replication gene is evolutionarily conserved, structurally stable, and might have consistent function.

Analysis of gene structure and conserved motifs

The CDPK genes were visualized for the gene structure and conserved motifs. The results showed that they have similar gene structure distribution patterns in the same subfamily. Most CDPKs of the first and the third subfamily have seven or eight exons, the CDPKs of the second subfamily have eight or nine exons, and the fourth subfamily has 12 exons. Most CDPKs of the family have 12 exons (Fig. 3). Generally, the structural conservation and similarity of the same type of genes are closely related to their evolutionary relationship, but there are some exceptions. GbCDPK38 has only five exons and GbCDPK61 has only three exons. The longest exon of most GbCDPKs is the first exon. However, there are still a small number of genes in the family members that have evolved significantly different structures. GbCDPK39 and GbCDPK81 are homologous genes from different sub-genomes, GbCDPK39 has a typical gene structure of this group, while the first exon of GbCDPK81 is relatively short. GbCDPK10 and GbCDPK11 are homologous genes of the same sub-genomes, but their gene structure is also different. GbCDPK10 has typical gene structure characteristics, but the first exon of GbCDPK11 is also shorter than that of GbCDPK10. This structural difference is likely to have formed randomly during evolution. The changed gene structure may lead to changes in gene splicing sites, thereby generating new transcripts or non-functional gene products, and ultimately increasing the number of members of the cotton GbCDPKs gene family.

Figure 3 Whole genome duplication distribution of CDPK genes in G. barbadense.

The MEME online software was used to conduct conservative analysis on the amino acid sequence of CDPK, and 10 conservative motifs were found (Fig. 4). These 10 conserved motifs exist in subgroups I, II, III and IV, but there are four genes (GbCDPK11, GbCDPK14, GbCDPK38, GbCDPK61) whose conservative motifs are distributed differently in the first subgroup. GbCDPK11, GbCDPK14 contains the same conserved motifs, without Motif 2, Motif 3, Motif 5. While GbCDPK38 has only three conserved motifs (Motif 8, Motif 9, Motif 10), GbCDPK61 has two conserved motifs (Motif 8, Motif 10). However, in the same subfamily, CDPK has a similar number, type, and spatial distribution of motifs, demonstrating that the functions of CDPK genes in the same subfamily are similar.

Figure 4 (A–C) Phylogenetic tree, conservative motif and gene structure in G. barbadense.

Analysis of promoter cis-elements

To explore the regulatory mechanisms of GbCDPK genes, the 1.5kb sequence upstream of the start condon of the GbCDPKs was selected to analyze the constitutions of cis-regulatory elements. It was found that all CDPK genes contained CAAT-box conserved elements and TATA-box. In addition to these conserved elements, there are four types of cis-regulatory elements in the promoter of CDPK gene: (1) Light regulatory elements, including Box 4, G-Box, TCCC-motif and AE-box, etc. (2) Plant growth and development regulatory elements, including CAT-box, GCN4_motif, O2-site, MBSI and circadian, etc. (3) Phytohormone response elements, including methyljasmonate response elements TGACG-motif and CGTCA-motif, (ABRE), gibberellin responsive element P-box, etc. (4) Stress response elements include LTR, MBS, ARE and mechanical damage response element W-box, etc.

Expression profile of different tissues and abiotic stress responses

To explore the possible biological functions of GbCDPKs, the transcriptome data of G. barbadense were used to analyze the expression characteristics of gene family in ovule and fiber at 0, 1, 3, 5, 10 and 20 days after anthesis, as well as in root, stem, leaf, receptacle, petal, stamen, calycle and sepals (Fig. 5A). The results showed that the CDPK gene has tissue-specific expression and spatio-temporal expression characteristics. According to the expression characteristics, the CDPK family can be divided into three types: the first group of genes are generally expressed less, but some genes were expressed in some or specific tissue. The second group was highly expressed in most tissues and developmental stages. The third group of genes are mainly expressed in vegetative organs or floral organs, but less expressed in ovule and fiber. Generally, homologous genes have similar expression patterns. Some homologous genes, such as GbCDPK13, GbCDPK56 and GbCDPK30, GbCDPK72 are highly expressed in the stamens. GbCDPK28, GbCDPK70; GbCDPK33, GbCDPK75 and GbCDPK38, GbCDPK80 are predominantly expressed in ovules. GbCDPK1, GbCDPK43 and GbCDPK25, GbCDPK67 are predominantly expressed in receptacles and stamens. The above results indicate that the CDPK gene family is widely involved in the growth and development of G. barbadense.

Figure 5 (A–B) Expression profile in different tissues and abiotic stress in GbCDPK.

CDPKs can regulate the response of plants to abiotic stress. The transcriptome data was used to analyze the expression characteristics of the CDPK gene of G. barbadense under low temperature, high temperature, drought (PEG6000), and salt treatment (Fig. 5B). According to the expression characteristics shown in response to abiotic stress, the CDPK gene family can be divided into four groups: the first group most genes are low expressed under adversity stress. The second group all genes is low expressed. The three groups are all highly expressed. The fourth group most genes are highly expressed. In addition, homologous genes have similar expression profiles. For example, some homologous genes GbCDPK31 and GbCDPK73 are highly expressed under all stresses.

Expression under drought stress

The transcript levels of CDPK genes in cluster 1 and 2 were decreased under drought. In order to further understand the expression of CDPK genes at transcriptional level in G. barbadense. qRT-PCR was performed using RNA isolated from drought stress (Fig. 6). After drought, the expression of GbCDPK4 showed a downward-increasing-decreasing trend, and the expression was the lowest at 6 h. GbCDPK19 and GbCDPK78 showed an overall upward-decreasing-increasing-decreasing trend. The former had the highest expression at 12 h, and the latter at 12 h. The expression level of GbCDPK23 was the highest at 3 h. The expression level of GbCDPK23 showed a trend of decrease-increasing-decreasing, and the expression level of 12 h was the highest. The expression levels of GbCDPK44 and GbCDPK66 showed a trend of increase-decrease-increasing, and the expression level of the former was the lowest at 6 h. The expression was highest at 24 h.

Figure 6 Abiotic stress expression of CDPK genes.

Silencing GbCDPKs compromises cotton resistance to drought stress

In order to further study the function of CDPKs drought tolerance in G. barbadense, we chose to design specific primers in the 3′-UTR region of CDPKs to construct VIGS vectors. The transcript levels of CDPK genes in cluster 3 and 4 were significantly increased under drought, with the highest peak observed all stage of the drought treatment. The empty vector (TRV-00) and the vector with target fragments (TRV-CDPKs) were activated and propagated. Then the cotton cotyledons were injected into the 10 days seedlings for gene silencing. After two weeks of injection, when the positive seedlings were albino (Fig. 7A), indicating success of the VIGS experiment. The observed gene expression patterns may reflect their functions. The interference efficiency of the target gene was detected by qPCR technology. The results show that the GbCDPKs genes (CDPK32, CDPK68, CDPK74, CDPK80 and CDPK83) in cotton plants injected with bacterial liquid containing the target gene fragments can indeed interfere effectively and specifically (Fig. 7B).

Figure 7 (A–I) Functional analysis of GbCDPKs in response to drought stress.

To determine the tolerance of GbCDPKs silenced plants under drought stress, leaf discs were made from the same parts of TRV-00 and TRV-GbCDPKs (TRV-CDPK32, TRV-CDPK68, TRV-CDPK74, TRV-CDPK80 and TRV-CDPK83). The leaves were incubated in 0 and 20% PEG6000 solution for 4 days. The results showed that the leaf chlorosis and wilting of gene silenced TRV-CDPK32, TRV-CDPK68, TRV-CDPK74, TRV-CDPK80 and TRV-CDPK83 plants were significantly more chlorosis and wilting than that of the control (Fig. 7C).

In addition, we measured several physiological indicators including RWC, REL, water loss rate and MDA, proline and soluble sugar content under drought treatment and normal growth conditions. As shown in Figs. 7D–7H, in normal conditions, relative water content, electrical conductivity, proline and soluble sugar content of TRV-00 were higher than TRV-GbCDPKs. However, the TRV-00 of MDA was lower than silencing. Under the drought stress, the relative water content, proline and soluble sugar content were decreased in TRV-CDPKs plants, compared with the TRV-00. The relative electrical conductivity, MDA of TRV-00 and TRV-CDPKs both increased. From Fig. 7I, TRV-CDPKs was significantly higher than TRV-00, indicating that the drought resistance of plants decreased remarkably after silenced. In conclusion, the drought tolerance of the plants is weaker after silenced.

Discussion

In this study, we have identified 84 presumptive CDPK genes in G. barbadense (AADD). The members of CDPKs family are significantly more than those in other species. There are 41 pairs of homologous gene pairs in the CDPK gene family, and 40 pairs are segmental duplication between the A/D chromosome pairs. Our analysis showed that segmental duplication is a predominant driving force that contributed to the expansion of GbCDPKs. The parallel evolution characteristics of homologous gene pairs in A-subgroup and D-subgroup are significant. They may come from the same ancestor, but the remaining one pair exists in the form of tandem replication (Liu et al., 2014). Generally, the amplification of CDPKs family members in angiosperms mainly depends on large-scale gene rearrangement, that is, whole genome or large fragment gene replication events (Hamel, Sheen & Séguin, 2014). This may explain the fact that there are more members of CDPKs family in G. barbadense than in Arabidopsis, rice and maize. Phylogenetic analysis indicated the GbCDPKs divided into four groups like Arabidopsis, and most CDPKs in the same group have similar gene structure, indicating the evolutionary and functional conservation of CDPKs. Generally, the orthologous genes have similar biological functions, while the paralogous genes have different biological functions. Therefore, the function of GbCDPK genes can be inferred from orthologous genes to provide scientific theoretical basis for subsequent functional studies (Thornton & DeSalle, 2000).

The specificity of signal transduction can be determined by the subcellular location of CDPK protein. It is predicted that GbCDPKs are in several cell compartments. It is also predicted that most GbCDPKs contain N-myristoylation and palmitoylation. The CDPK genes of N- myristoylation motif tends to be in the plasma membrane (Martín & Busconi, 2000; Lu & Hrabak, 2013) The myristoylation may be a part of the main signal transduction process that guides proteins to membrane binding sites. In addition, it is reported that another lipid-modified palmitoylation is necessary for the stability of membrane association (Asai et al., 2013). In this study, 40 genes have N-myristoylation and 75 genes have palmitoylation motifs. However, the association of CDPKs with membranes is complex and may be affected by other motifs. For example, in wheat (Triticum aestivum), TaCPK3 and TaCPK15 without myristoylation regions are associated with the membrane (Li et al., 2008), while in Maize (Zea mays), N-myristoylation patterns were found in ZmCPK1 of the cytoplasm and nucleus (Wang & Shao, 2013).

Other studies have shown that the subcellular location of CDPKs can be limited in a single compartment or widely distributed in the whole cell. CDPKs have been found in plasma membrane, cytoplasm, nucleus, endoplasmic reticulum, mitochondria, chloroplast, oleosome, peroxisome and Golgi apparatus (Asano et al., 2012a). To explore the function of proteins in plant, subcellular localization is one of the essential research. Ca2+ pumps are known to be widely distributed on membranes (Tanaka et al., 2004), CDPK can rapidly respond to transient changes of Ca2+ signal in plants and specifically recognize substrates after phosphorylation. Various physiological responses were further triggered in signaling transduction cascades, which finally regulate plant growth and development, and in response to multiple stresses (Shi et al., 2018). We found that StCDPK7 protein with auto-phosphorylation activity could locate in the nucleus, membranes and interact with PAL protein, suggesting that the auto-phosphorylation activity of StCDPK7 protein played important roles in protein location and protein-protein interactions (Fantino et al., 2017). SiCDPK24 protein was proven to have a role in auto-phosphorylation, and may be important for its function. SiCDPK24-GFP localized to the plasma membrane and the nucleus (Yu et al., 2018). Therefore, the subcellular localization of CDPK genes are closely related to drought stress response.

CDPKs activated by transient changes in Ca2+ concentration in plants is essential for various biological processes (such as plant growth, development, regeneration, defense against biotic and abiotic stresses, etc.) (Asano et al., 2012a; Ranty et al., 2016). It has been reported that CDPKs are widely distributed in plant tissues, even in fruit, pollen tubes, germs, and guard cells (Chang et al., 1995; Zhou, Fu & Yang, 2009). The CDPKs protein family is huge, and its functions are also diverse (Asano et al., 2012b). We use the transcriptome data to evaluate the expression pattern of cotton CDPKs in 17 different tissues. The results of tissue expression distribution indicated that the gene family of cotton CDPKs was widely distributed, implying that its function is diverse. CDPKs expressed in many tissues may participate in a variety of biological processes, some members specifically expressed in stamens may participate in reproductive development, and some may participate in stress response process. The specific genes involved in which biological process need to be further studied.

With the change of global ecological environment, drought has become one of the main environmental factors limiting crop quality and yield. In other plants, CDPKs was associated with drought tolerance, such as StCDPK3 and StCDPK23 and PtrCDPK10 are all involved in drought stress (Bi et al., 2021; Meng et al., 2020). However, few people have studied the function of CDPKs in G. barbadense.

To further study the function of drought tolerance in GbCDPKs, VIGS technology was used to silence candidate CDPK genes to construct transient silence materials (Senthil-Kumar & Mysore, 2014). Our analysis indicated that CDPKs silencing plants were more sensitive to drought stress than control plants. The results of the study showed that the drought resistance ability of CDPKs silencing was significantly reduced. The content of proline and the MDA was higher than the control. On the contrary, the relative water content of the silenced plants was lower than that in the controls under drought stress. To maintain the stability of protoplast colloid, the proline is accumulated to increase the abiotic pressure of plant cells under drought stress (Sallam et al., 2019). In this study, drought stress induced accumulation of proline in cotton, and thereby enhanced abiotic stress adjustment ability. The drought sensitive phenotype of silencing plants to drought stress indicates that CDPKs play a regulatory role in improving cotton drought tolerance, but whether they participate in the same drought tolerance regulatory mechanism is still unknown, and further research is needed to prove it.

Conclusions

Eight-four CDPK genes were identified in G. barbadense. They all have the EF-hands structure and Ser/Thr protein kinase domain, none of which include the transmembrane structure. They were found to express in different organelles by subcellular localization. According to the phylogenetic tree, all CDPKs were divided into four groups, which were distributed on 13 pairs of chromosomes. The CDPKs are distributed in 13 tissues. In response to abiotic stress, it showed different expression patterns. The silence of some CDPKs severely destroyed the basic resistance of cotton to drought stress, indicating that they were involved in the resistance mechanism. This study deepens our understanding of the function of CDPK and provide important insights for further research on the molecular mechanism of G. barbadense to drought stress and to provide candidate target genes that improve drought adaptation.

Supplemental Information

Appendix S1A RT-PCR and qPCR reaction systems and procedures

Click here for additional data file.

Appendix S2B Primers used in the experiment

Click here for additional data file.

Table S1 Molecular structure physiological and biochemical characteristics of CDPK family in G.barbadense

Click here for additional data file.

Table S2 The Ka/Ks ratios for duplicate GbCDPK genes

Click here for additional data file.

Table S3 Analysis of cis-regulatory elements in the promoters of each gene

Click here for additional data file.

Supplemental Information 6 expression profile in different tissues raw data

Click here for additional data file.

Supplemental Information 7 Expression profile under abiotic stress raw data

Click here for additional data file.

Supplemental Information 8 Abiotic stress expression of CDPK genes raw data

Click here for additional data file.

Supplemental Information 9 Relative expression after silence raw data

Click here for additional data file.

Supplemental Information 10 Physiological indicators raw data

Click here for additional data file.

We would like to express gratitude to Prof. Zhang Tianzhen and Assoc. Prof. Fang Lei from Zhejiang University for providing G. barbadense transcriptome data.

Abbreviations

CDPKs Calcium dependent protein kinases

VIGS virus-induced gene silencing

qRT-PCR quantitative reverse transcription PCR

G. barbadense Gossypium barbadense

RWC Relative water content

REL Relative electrical conductivity leakage

MDA Malondialdehyde

Additional Information and Declarations

Competing Interests

Author Contributions

Data Availability

The authors declare there are no competing interests.

Guangzhen Shi performed the experiments, analyzed the data, prepared figures and/or tables, authored or reviewed drafts of the paper, and approved the final draft.

Xinxia Zhu conceived and designed the experiments, authored or reviewed drafts of the paper, and approved the final draft.

The following information was supplied regarding data availability:

The raw measurements are available in the Supplementary File.

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
