# Peer review of "Genome-wide identification and functional characterization of CDPK gene family reveal their involvement in response to drought stress in Gossypium barbadense"

_PeerJ, doi:10.7717/peerj.12883_

## Round 0.1 · original submission · Minor Revisions

Please submit the raw transcriptomics data in the public database and provide the accession number of data. Please also provide the detailed methods for RWC, RE,MDA, PROLINE, SOLUBLE SUGAR AND WATER LOSS RATEE in the material and methods section.

Reviewer 1 ·

Basic reporting

This manuscript present clear and unambiguous conclusions. I commend the authors for their extensive data set, however, the authors should provide the raw transcriptomics data in the public database (i.e. NCBI), so that the original data are available to the public.

Experimental design

The methods are thorough.

Validity of the findings

The manuscript presents the results which are well presented and appropriately described. Conclusions are appropriate, and supported by the data.

Additional comments

Shi et al presented a genome-wide identification and characterization of 84 CDPK genes in G. barbadense. In this study, the authors used bioinformatics to identify the members of CDPK gene family and predict the structure, physiological and biochemical features of the protein. Further, the authors analyzed the evolution and classification, and the expression of the CDPK genes. Finally, the authors demonstrated that these CDPKs are associated with the drought.

The manuscript presents the results which are well presented and appropriately described. The methods are thorough. Conclusions are appropriate, and supported by the data. The manuscript is overall sound, and below I have highlighted a few minor concerns.

1.With regarding to data availability: the authors should provide the raw transcriptomics data in the public database (i.e. NCBI), so that the original data are available to the public.

2. Line 29: it should be CDPKs for the short name Calcium dependent protein kinases. The author should check other places in the manuscript for the correction.

Reviewer 2 ·

Basic reporting

no comment

Experimental design

no comment

Validity of the findings

no comment

Additional comments

Calcium dependent protein kinases (CDPKs) are a class of important calcium signal sensing response proteins, which play an important regulatory role in response to abiotic stress. Shi et al identified the CDPK family genes in sea-island cotton genome and performed a series of bioinformatics analysis, and curried out qRT-PCR for some genes under different stresses and further selected some genes for VIGS analysis of drought tolerance. These findings provide a solid foundation for the development of future molecular mechanism in sea-island cotton. This study is very carefully done and the paper is organized very well, except a few minor mistakes.
Minor
1. Some English spellings, such as line 59, 67-68,89,165,231,246,249,303,341,379.
2. Please cite this paper and compare with your result in the manuscript. Liu W, Li W, He Q, Daud MK, Chen J, Zhu S. Genome-wide survey and expression analysis of calcium-dependent protein kinase in Gossypium raimondii. PLoS One. 2014 Jun 2;9(6):e98189
3. Please provide the detailed methods for RWC, RE,MDA, PROLINE, SOLUBLE SUGAR AND WATER LOSS RATEE in the material and methods section.

Reviewer 3 ·

Basic reporting

..

Experimental design

.

Validity of the findings

.

Additional comments

This manuscript conduct the CDPK gene family characteristics analysis and also analyze the potential function of some of them respond to drought stress. The manuscript was well written and the story is appealing. However, there are several questions need to be clarified before the publication.
1) The figures are not clear enough, I wonder this is because the figure was inserted into the manuscript, so please check the figure quality
2) Can you add the relationship analysis about the subcellular location and the response to drought stress of some CDPK gene in the DISCUSSION Part.

Minor questions:
1) Some minor mistakes need to be noted: such as the figure 7 legend, t-test, the first t should be italic; P value, the P should be italic.
2) Line 38 sea should be Sea
3) Line 72-73 should be rewritten
4) N-myristoylation should be N, the N should be italic
5) Line 267 CDPK should be italic

---

## Round 0.2 · Minor Revisions

The manuscript has been improved. I have received comments from the third Reviewer and there are several questions that need to be clarified before publication.

1) The figures are not clear enough and suitable for publication, I wonder if this is because the figure was inserted into the manuscript, so please check the figure quality.
2) Can you add the relationship analysis about the subcellular location and the response to drought stress of some CDPK gene in the DISCUSSION Part?

Minor questions:
1) Some minor mistakes need to be noted: such as the figure 7 legend, t-test, the first t should be italic; P value, the P should be italic.
2) Line 38 sea should be Sea
3) Line 72-73 should be rewritten
4) N-myristoylation should be N, the N should be italic
5) Line 267 CDPK should be italic

---

## Round 0.3 · accepted · Accept

The Reviewers are satisfied with the revisions. Thus we can now proceed with acceptance of the manuscript in its current form.

Reviewer 2 ·

Basic reporting

no comment

Experimental design

no comment

Validity of the findings

no comment

Additional comments

The authors have answered all my concerns, I suggest accept it in the present form.

Reviewer 3 ·

Basic reporting

Clear and unambiguous, professional English used throughout.

Literature references, sufficient field background/context provided.

Professional article structure, figures, tables. Raw data shared.





Self-contained with relevant results to hypotheses.

Experimental design

Clear and unambiguous, professional English used throughout.

Literature references, sufficient field background/context provided.

Professional article structure, figures, tables. Raw data shared.





Self-contained with relevant results to hypotheses.

Validity of the findings

Impact and novelty not assessed. Meaningful replication encouraged where rationale & benefit to literature is clearly stated.

All underlying data have been provided; they are robust, statistically sound, & controlled.

Conclusions are well stated, linked to original research question & limited to supporting results.